# Haplotype-Phased Chromosome-Level Genome Assembly of *Cryptoporus qinlingensis*, a Typical Traditional Chinese Medicine Fungus

**DOI:** 10.3390/jof11020163

**Published:** 2025-02-19

**Authors:** Yu Song, Ming Zhang, Yu-Ying Liu, Minglei Li, Xiuchao Xie, Jianzhao Qi

**Affiliations:** 1Shaanxi Province Key Laboratory of Bio-Resources, Qinba State Key Laboratory of Biological Resources and Ecological Environment (Incubation), School of Biological Science and Engineering, Shaanxi University of Technology, Hanzhong 723000, China; 2Shaanxi Key Laboratory of Natural Products & Chemical Biology, College of Chemistry & Pharmacy, Northwest A&F University, Yangling 712100, China; 3Center of Edible Fungi, Northwest A&F University, Yangling 712100, China

**Keywords:** genome phasing, secondary metabolic potential, *Cryptoporus qinlingensis*, medicinal fungi

## Abstract

This study presents the first comprehensive genomic analysis of *Cryptoporus qinlingensis*, a classical folk medicine and newly identified macrofungus from the Qinling Mountains. Utilizing advanced sequencing technologies, including PacBio HiFi and Hi-C, we achieved a high-quality chromosome-level genome assembly. The genome, sized at 39.1 Mb, exhibits a heterozygosity of 0.21% and contains 21.2% repetitive sequences. Phylogenetic analysis revealed a recent divergence of *C. qinlingensis* from *Dichomitus squalens* approximately 212.26 million years ago (MYA), highlighting the rapid diversification within the Polyporaceae family. Comparative genomic studies indicate significant gene family contraction in *C. qinlingensis*, suggesting evolutionary adaptations. The identification of a tetrapolar mating system, along with the analysis of CAZymes and P450 genes, underscores the genomic complexity and ecological adaptability of this species. Furthermore, the discovery of 30 biosynthetic gene clusters (BGCs) related to secondary metabolites, including polyketide synthase (PKS), non-ribosomal peptide synthetase (NRPS), and terpene synthesis enzymes, opens new avenues for exploring bioactive compounds with potential medicinal applications. This research not only enriches our understanding of the *Cryptoporus* genus but also provides a valuable foundation for future studies aiming to harness the therapeutic potential of *C. qinlingensis* and to further explore its ecological and evolutionary significance.

## 1. Introduction

*Cryptoporus*, a genus within the family Polyporaceae (Basidiomycota), includes species such as *Cryptoporus volvatus* and *Cryptoporus sinensis*, which are recognized as key members [1]. The fruiting bodies of these fungi are typically flattened, spherical or globular, with a smooth surface and a deep eggshell color, and are often found on decaying wood or fallen coniferous trees [2]. In China, they are widespread in provinces such as Yunnan, Sichuan, Zhejiang and Hainan, where their fruiting bodies develop rapidly, especially during the rainy spell in early summer [2]. As a medicinal fungus of significant value, the fruiting bodies of *Cryptoporus* have been used in traditional Chinese medicine for centuries to treat asthma and bronchitis [2]. Recently, a new species of the genus *Cryptoporus* was discovered on the trunk of a Masson pine (*Pinus massoniana* Lamb.) in the Qinling Mountains, northwestern China. This new species was identified through phylogenetic analysis based on multi-molecular marker sequences and detailed morphological characteristics and has been named *Cryptoporus qinlingensis* Y. Wang, X.C. Xie & Y. Song [3]. It is also the third species of the genus to be described.

Chemical investigations of *Cryptoporus* have uncovered a range of bioactive constituents, including sesquiterpenoids [4,5,6,7], ergosterol, and its peroxides [8,9,10], as well as polysaccharides [11]. Notably, sesquiterpenoid compounds such as cryptoporic acid derivatives and isocryptoporic acid exhibit remarkable biological activities [1,4,5,6,7]. Ergosterol peroxide has been shown to be effective in anti-tumor and anti-inflammatory activities [8,9,10,12]. Polysaccharide components have been identified to play a crucial role in enhancing immune response and exhibiting antitumor properties [13,14]. Analysis of the volatile oil components of *Cryptoporus* has led to the identification of multiple compounds, including palmitic acid and oleic acid, which have shown anti-inflammatory and analgesic effects [15]. The pharmacological effects of *Cryptoporus* are primarily reflected in its anti-inflammatory, asthma-relieving, and anti-tumor capabilities. Studies have indicated that polysaccharides from *Cryptoporus* significantly suppress eosinophils and inflammatory cells in airway inflammation models, thereby improving lung function [14]. Furthermore, *Cryptoporus* extracts have shown a strong inhibitory effect on porcine reproductive and respiratory syndrome viral infection [16,17]. Sesquiterpenoid components from *Cryptoporus*, identified through molecular docking techniques, are likely to bind with the N-terminal protein sequence 1 UNQ of protein kinase B, inducing acetylation at the Lys-14 position, which may affect the proliferation of tumor cells [18]. Additionally, research on the carbon isotope fractionation in *Cryptoporus* has revealed growth-dependent fractionation of stable carbon isotopes during its growth process, offering new insights into the role of fungi in ecosystems [19].

As of 4 January 2025, the NCBI GenBank database has included nearly 3000 Basidiomycota genomes. However, the number of genome records for species in the family Polyporaceae is less than 80. It is particularly noteworthy that no species in the genus *Cryptoporus*, which is an important group in this family with both medicinal value and ecological rarity, has completed whole-genome sequencing. This knowledge gap severely restricts the genetic research and medicinal development of fungi in this genus. To elucidate the genomic data, genetic traits, evolutionary status, CAZymes, mating type loci, cytochrome P450, and biosynthetic genes of active components of *C. qinlingensis*, this project undertakes high-quality genome sequencing of a *C. qinlingensis* derived from folk medicinal practices in Hanzhong city, Shaanxi Province, China. This endeavor aims to unlock the genomic traits of *C. qinlingensis*, providing a foundation for further research into its medicinal properties. By deciphering its genomic blueprint, we can better understand the mechanisms underlying its pharmacological effects, trace its evolutionary lineage, and identify the specific genetic loci responsible for mating. Additionally, the identification of biosynthetic pathways for its bioactive components will pave the way for potential enhancement of their production, either through targeted breeding or biotechnological approaches. This comprehensive genomic analysis will not only enrich our knowledge of *Cryptoporus* but also contribute to the development of new therapeutic agents and strategies in medicine.

## 2. Materials and Methods

### 2.1. Fungal Material and Nucleic Acid Extraction

The fruiting bodies of the *C. qinlingensis* (Figure 1A) were initially collected from a living Masson pine in the mountainous region of Hanzhong City, Shaanxi Province, China, an area that is part of the Qinling mountain range, as reported previously [3]. Cultivable mycelium was isolated from fresh fruiting bodies of *Cryptoporus* by tissue separation. This mycelium is maintained as a fungal strain resource in the form of slant cultures at the Shaanxi Provincial Engineering Research Center and the Shaanxi Key Laboratory of Natural Products and Chemical Biology and with the accession number SNUT. *Cryptoporus qinlingensis* SNUT was cultured on potato dextrose agar (PDA, Difco, BD, Sparks, MD, USA) for 10 days at 25 °C to obtain a sufficient amount of mycelium. Genomic DNA samples were extracted from the mycelium using the Fungal DNA Mini Kit (Omega, Norcross, GA, USA) according to the manufacturer’s instructions. The purified DNA samples were then subjected to quality control. Genomic DNA, using a high purity DNA sample with an OD260/280 ratio of 1.8 to 2.0 and a concentration greater than 20 mg/mL, was used to construct a fragment library using the TBS-380 fluorimeter (Turner BioSystems Inc., Sunnyvale, CA, USA). RNA extraction from the mycelium was performed using RNA isolater Total RNA Extraction Reagent (Vazyme, Nanjing, China) according to its instructions.

### 2.2. Genome Sequencing, De Novo Assembly, Annotation, and Visualization

#### 2.2.1. NGS, PacBio, and RNA-Seq Library Construction and Sequencing

For NGS paired-end sequencing, a minimum of 1 μg of genomic DNA is required to construct the sequencing library. Following the standard protocol for genomic DNA library preparation, a paired-end library with an insert size of approximately 400 base pairs is fabricated. The purified genomic DNA is sheared to the desired size using a Covaris sonicator (Covaris, Woburn, MA, USA), and then T4 DNA polymerase is employed to create blunt ends. After adding an “A” base to the 3′ end of the phosphorylated blunt DNA fragments, adapters are ligated to the ends of the DNA fragments. The required fragments are purified through gel electrophoresis, selectively enriched, and subsequently amplified by PCR. During the PCR stage, index tags may be incorporated into the adapters as needed, and the library is subjected to quality testing. Ultimately, qualified NGS paired-end libraries will be used for sequencing (150 bp × 2) to ensure that high-quality, well-prepared samples are used for the sequencing process.

For PacBio sequencing, the standard protocol is used to generate and sequence a 20 kb insert size whole-genome shotgun library on the Sequel II platform. Equal aliquots of 8 μg DNA were sheared in a Covaris g-TUBE (Covaris, Woburn, MA, USA) and then centrifuged for 60 s at 6000 rpm using an Eppendorf 5424 centrifuge (Eppendorf, New York, NY, USA). Following the manufacturer’s recommendations (Pacific Biosciences, Menlo Park, CA, USA), DNA fragments were purified, end-repaired and ligated using SMRTbell sequencing adapters. The sequencing library is then purified using 0.45 volumes of Agencourt AMPure XP beads (Beckman Coulter Genomics, Danvers, MA, USA) according to the manufacturer’s guidelines to ensure library quality and integrity for optimal sequencing performance.

To facilitate the annotation of transcriptomes within the genome, a strand-specific RNA-seq library with an insert size of 350 base pairs was prepared using the NEBNext^®^ Ultra^TM^ RNA Library Prep Kit for Illumina^®^ (New England Biolabs, Ipswich, MA, USA). This library was then sequenced on an NGS platform to generate paired-end reads of 150 base pairs each. This process ensures comprehensive and strand-specific transcriptome analysis, providing valuable insights into gene expression profiles.

#### 2.2.2. De Novo Assembly, Haplotype Phasing, and Hi-C Scaffolding

Raw paired-end reads were trimmed to remove adapters and poor-quality bases using Trimmomatic v0.39 (https://github.com/timflutre/trimmomatic, accessed on 9 October 2024) after quality control with FastQC (https://github.com/s-andrews/FastQC, accessed on 13 October 2024). The reads were filtered using a sliding window of size four, with an average Phred score threshold of 20 within the window. The trimmed reads were then used for further analysis. The de novo genome assembly integrated sequencing data from a variety of technologies, including whole-genome shotgun sequencing with NGS and PacBio Sequel and Hi-C reads, followed by polishing with PacBio reads and NGS reads. The assembly of PacBio HiFi long reads was performed using Hifiasm (https://github.com/chhylp123/hifiasm, accessed on 13 October 2024), which is capable of error correction that recognizes haplotypes. Self-correction was performed, which allowed us to correct all input PacBio reads. The corrected reads were then imported into the assembly process. The chromosomal assembly was constructed using the ALLHIC program v 0.9.14 (https://github.com/tanghaibao/allhic, accessed on 13 October 2024), which was designed to scaffold polyploid genomes and is also applicable to diploid genomes.

Briefly, the chromosome assembly consists of the following steps: (1) Hi-C reads are mapped to the draft assembly using bwa v0.7.17 (https://github.com/bwa-mem2/bwa-mem2/issues/262, accessed on 13 October 2024) and poor quality read pairings are removed from the resulting BAM file using PreprocessSAMs.pl from LACHESIS (https://github.com/shendurelab/LACHESIS, accessed on 13 October 2024). (2) All contigs are then partitioned into separate clusters based on the number of links between them. (3) The highest scoring order and orientation for each cluster are reconstructed. (4) The Hi-C-based chromosome-level assembly is built and the heatmap is plotted to illustrate chromosome contacts. To achieve a phased haplotype assembly, HaploMerger2 (https://github.com/mapleforest/HaploMerger2, accessed on 13 October 2024) was utilized to deduce the haploid subassemblies from a highly heterozygous diploid genome. The heterozygosity was quantified using Genomescope (https://github.com/schatzlab/genomescope, accessed on 13 October 2024).

#### 2.2.3. Gene Prediction and Genome Annotation

Gene prediction and annotation were primarily conducted using BRAKER v2.1.4 (https://github.com/Gaius-Augustus/BRAKER, accessed on 13 October 2024), a tool designed for the annotation of eukaryotic genomes. The model was trained with GeneMark-EX and employed AUGUSTUS v3.3.3 (https://github.com/Gaius-Augustus/Augustus, accessed on 13 October 2024) for the prediction of open reading frames (ORFs). AUGUSTUS genome annotation training was performed using a collection of fungal genomes such as *D. squalens* and *Inonotus hispidus*. The training set was fine-tuned to optimize the prediction accuracy of the *C. qinlingensis* genome to ensure the reliability of gene predictions. For the prediction and classification of non-coding RNAs, INFERNAL v1.1.2 (https://github.com/EddyRivasLab/infernal, accessed on 13 October 2024) was utilized in conjunction with the Rfam 14.6 database. Furthermore, the gene products were annotated with the help of non-redundant protein sequences from NCBI, Swiss-Prot, COG, GO, KOG, and KEGG databases.

#### 2.2.4. Genomic Circular Map

The circular genomic map was constructed using McscanX (https://github.com/wyp1125/MCScanX, accessed on 13 October 2024), which represents circular genomes in a graphical map, resulting in base composition plots, sequence features, and analyses such as GCskew, GC content, and collinearity.

#### 2.2.5. Comparative Genomics Analysis

A comparative genomic study was performed utilizing OrthoFinder v2.5.5, with the software configured to employ specific parameters: the diamond tool for sequence searching, msa for multiple sequence alignment, FastTree 2 (https://github.com/morgannprice/fasttree, accessed on 13 October 2024) for phylogenetic tree construction, and a thread count of 128. The outcomes of this comparative genomic investigation were then elegantly depicted through the use of jVenn (http://jvenn.toulouse.inra.fr/app/index.html, accessed on 13 October 2024) for visual representation. To ascertain the ratio of synonymous to nonsynonymous substitution rates, denoted as Ks to Ka, between *Cryptoporus* species and *D. squalens*, we undertook comprehensive genomic replication studies. These were executed with the aid of wgd v1.1.2 (https://github.com/arzwa/wgd, accessed on 13 October 2024) and Para AT v2.0 (https://ngdc.cncb.ac.cn/tools/paraat, accessed on 15 October 2024). This approach facilitated a detailed examination of the evolutionary pressures at play within these fungal species.

#### 2.2.6. Repeat Sequence Identification

Using RepeatModeler v2.0.2 (https://github.com/Dfam-consortium/RepeatModeler, accessed on 13 October 2024) and RepeatMasker v4.1.5 (https://www.repeatmasker.org/RepeatMasker/, accessed on 13 October 2024), we predicted four types of transposable elements (TEs) in the genome, namely long terminal repeats (LTRs), long interspersed elements (LINEs), short interspersed elements (SINEs), and DNA transposable elements (DNA-TEs). First, a custom repeat library containing *C. qinlingensis* was constructed using RepeatModeler and merged with the Repbase library. Subsequently, RepeatMasker was used to annotate the repetitive sequences in the genome.

### 2.3. Phylogenomic Analysis and Gene Family Variation Analysis

A phylogenetic examination was undertaken to delineate the evolutionary ties amongst *Cryptoporus* species and a selection of 30 archetypal Basidiomycetes. The identification of single-copy orthologous genes was accomplished employing OrthoFinder v2.5.5 (https://github.com/davidemms/OrthoFinder, accessed on 13 October 2024), with the command-line parameters tailored to “-S diamond -M msa -T raxml-ng”. Chronological estimates of divergence for 244 single-copy orthologous gene sequences from a spectrum of 31 strains were deduced through the application of MCMCTree, nestled within the PAML 4.9 e framework, accessible at the University College London’s Abacus software suite. The temporal dispersion of several relatively recent ancestral assemblages was scrutinized via TIMETREE 5 (http://www.timetree.org, accessed on 13 October 2024), with a focus on the comparative analysis of *Phanerochaete sordida* vs *Phanerochaete carnosa* dating from 2.27 to 23.82 MYAs, and *Marasmius oreades* versus *Lentinula edodes* ranging from 22.6 to 135.5 MYA. The evolutionary affinities among these species were elegantly depicted through the use of FigTree v 1.4.3. (http://tree.bio.ed.ac.uk/software/figtree/, accessed on 13 October 2024). For the evaluation of gene family dynamics, including expansions and contractions, the CAFÉ 4.2.1 toolkit (https://github.com/hahnlab/CAFE, accessed on 13 October 2024) was leveraged, engaging the identified orthologous gene families. The execution of this analysis was guided by a set of parameters: “--cores 30 --fixed_lambda 0.0001”, reflecting a meticulous approach to the computational demands of the task.

### 2.4. CAZy Family and Cytochrome P450 Analyses

Genes responsible for the production of enzymes that engage with carbohydrates, known as carbohydrate-active enzymes or CAZymes, in saprophytic fungi were meticulously annotated and categorized utilizing the comprehensive CAZy database. This was achieved with the application of the HMMER package, specifically version 3.2.1, employing stringent filter parameters that ensured an E-value threshold of less than 1e^−18^ and a coverage extent surpassing 0.35. The visual representation of the CAZyme analysis for white-rot Basidiomycota was masterfully crafted into a bubble plot, facilitated by the Complex Heatmap package within the Rstudio environment, version 4.20.

Cytochrome P450s, a class of proteins pivotal in drug metabolism and detoxification processes, were identified and examined through the use of Hmmer in conjunction with Diamond V2.9.0, applying an E value filter of less than e^−5^ to pinpoint the target protein sequences. For the cluster analysis, a curated selection of reference CYP sequences was sourced from the authoritative Fungal Cytochrome P450 Database.

A cohort of 77 cytochrome P450 proteins, derived from *C. qinlingensis* SNUT and a select group of other Basidiomycota species as listed in the database, were subjected to phylogenetic tree analysis, resulting in a clear and distinct classification. The construction of a robust maximum likelihood phylogenetic tree was accomplished with IQ-tree 2.2.3, employing parameters tailored for accuracy and reliability: “-m Q. pfam + R10 (best-fit model) -bb 1000 -alrt 1000 -abayes -nt AUTO.”

### 2.5. BGC Analysis and Visualization

The identification of BGCs was successfully executed using the fungal adaptation of antiSMASH 7.0 (https://fungismash.secondarymetabolites.org/#!/start, accessed on 4 October 2024). Phylogenetic tree-based clustering was performed utilizing IQtree 2.2.3, with the application of specific parameters “-m MFP -bb 1000 -alrt 1000 -abayes -nt AUTO” for robust analysis. For an in-depth examination of multi-domain enzymes, such as non-ribosomal peptide synthetases (NRPSs) and polyketide synthases (PKSs), the analytical capabilities of Synthaser (https://github.com/gamcil/synthaser, accessed on 13 October 2024) were engaged to dissect their domain architecture. This includes domains like adenylation (A), acyl carrier protein (ACP), acyltransferase (AT), thiolation (T), thioesterase (TE), condensation (C), β-ketoacyl synthetase (KS), product template (PT), acyl carrier protein transacylase (SAT), and thioester reductase (TR). The comparative assessment of homology and similarity between two BGCs was conducted with Clinker (https://github.com/gamcil/clinker, accessed on 13 October 2024). The visualization of these comparative findings was elegantly rendered through clustermap.js, an integrated tool within Clinker designed to produce comprehensive BGC comparison plots.

### 2.6. Data Availability

The final genome assembly results and associated data of *C. qinlingensis* SNUT have been submitted to NCBI under BioProject PRJNA1082898 and BioSample SAMN40220998, respectively. The genome assembly data that support the findings of this study are available from the corresponding author, [J. Qi.], upon reasonable request.

## 3. Results

### 3.1. Chromosome-Level Genome Assembly and Haplotype-Phasing of Cryptoporus

A total of 1.54 Gbp of PacBio HiFi reads (~44.8 × coverage) from single-molecule real-time sequences on the PacBio Sequel II (Appendix A) and 6.2 Gbp of Hi-C clean data and 6.4 Gbp (~184.3 × coverage) of de novo clean data (~179.8× coverage) from Illumina NovaSeq (Appendix A) were used to assemble the genome of *C. qinlingensis* SNUT. Genomic evaluation based on K-mer statistics showed that the strain SNUT had a genome size of 39.1 M, with 0.21% heterozygosity and 21.2% repetitive sequences (Appendix A, Appendix A). Two sets of haplotype-resolved, phased contig assemblies were generated using Hifiasm based on Pacbio HiFi long-reads, including the 34,555,381 bp haplotype A with 2,645,904 bp N50 and the 34,378,451 bp haplotype B with 2,587,211 bp N50 (Figure 1B,C, Appendix A). The assessment of GC-depth distributions with a significant Poisson distribution indicates high-quality genome assembly (Appendix A). After Hi-C-assisted assembly by ALLHIC, 34 Mbp of genome sequence was mapped to 13 chromosomes (Figure 1D, Appendix A) and a contig, the latter of which was subsequently found to be a circular mitochondrial genome. The 13 chromosomes ranged in length from 1,504,627 bp to 4,017,150 bp (Appendix A), totaling 99.75% of haplotype A. 97.6% of the BUSCOs (including 96.3% of the single-copy BUSCOs) indicated that the genome has good assembly completeness (Appendix A).

Subsequently, we conducted a comprehensive analysis to identify the telomeric motif, (TTAGGG)n, within the chromosomal structure (Figure 1E,F, Appendix A). In haplotype A, the telomeric motif was consistently detected at the termini of nine chromosomes, specifically A1, A4, A6 through A11, and A13, and was also observed at a single terminus of six additional chromosomes, namely A2, A3, and A5. In contrast, haplotype B exhibited the motif at both ends of ten chromosomes, encompassing B1, B4 through B11, and B13, and at a single end of three chromosomes, which are B2, B3, and B13. The telomeric sequence length varied significantly, extending from 64 bp (representing 14 repeats) to 258 base pairs (corresponding to 43 repeats). This variation underscores the heterogeneity in telomeric sequence lengths across different chromosomal regions and haplotypes.

### 3.2. Genome Annotation and Comparative Genome Analysis

Functional gene prediction of the genome was performed using a combination of de novo prediction, homologous protein comparison, and transcriptome data. There were 7750 and 7779 protein-coding genes predicted from the genomes of haplotypes A and B, with 33.66% and 34.00% of their respective genomic length, respectively (Appendix A).

Annotation of protein-coding genes based on five databases, NR, COG, Swiss-Prot, KEGG, and GO, yielded 7740 proteins in A haplotype and 7768 proteins in B haplotype (Appendix A, Appendix A). The results of the NR database annotation found that 7475 (96.58%) annotated genes matched A haplotype, and 7505 (96.61%) annotated genes matched B haplotype (Appendix A). Cluster annotation of orthologous groups based on the COG database identified 4674 genes in A haplotype, and 4672 genes in B haplotype (Appendix A). Subgroup S with unknown function contained the highest number of genes in the KOG annotation (Appendix A). Based on the Swiss-Prot database, 4413 genes were annotated in the genomes of both haplotypes A and B (Appendix A). According to the KEGG database, 3117 genes were identified in A haplotype as being involved in five types of pathways, and 3120 genes in B haplotype, with the largest number of genes involved in global and overview maps (Appendix A). Biological process constituted the main group among the 3016 genes in A haplotype and 3020 genes in B haplotype annotated by the functional classification of the GO database (Appendix A). In addition, 305 tRNAs, 16 rRNAs, and 16 snRNAs are predicted in the diploid assembly (Appendix A). Using the de novo library of RepeatMasker and the genome-trained library of RepeatModeler, we detected repeats accounting for 26.64% of the genome (both haplotypes), which is significantly higher than that of I. hispidus [20], a fellow member of the Polyporaceae family.

In order to delve deeper into the genomic characteristics of Cryptoporus, an investigation was conducted into the interhaplotype gene content and variation. The two haplotypes exhibited a comparable degree of functional annotation, with approximately 99% of proteins possessing at least one functional annotation. Utilizing Orthofinder, a total of 7191 common orthogroups were identified between the two haplotypes, along with 18 unique genes specific to haplotype A and 19 unique genes specific to haplotype B. Subsequently, for comparative analysis, *D. squalens*, a closely related species with a genome available, was employed. Orthofinder identified a total of 6224 common orthogroups across *Dichomitus squalens* and both haplotypes, as well as 6240 common orthogroups between haplotype A and haplotype B (Figure 1G). To further explore genetic variations, a genome-wide duplication analysis was conducted using rates of nonsynonymous (Ka) and synonymous (Ks) substitutions. The resulting Ka/Ks ratio curves for Cryptoporus species and *D. squalens* revealed marked differences, indicating distinct levels of evolutionary selection pressure. In the case of the two haplotyes, the incomplete overlap curves indicate that they have not experienced exactly the same selection for evolutionary pressures either (Figure 1H).

### 3.3. Phylogenetic and Gene Family Variation Analysis

To elucidate the evolutionary history of Cryptoporus species, we reconstructed a phylogenetic tree and estimated divergence times for a curated selection of 31 Basidiomycetes species dominated by Polyporales. This analysis was performed using 244 con-served single-copy orthologous proteins, with Ustilago maydis as an outgroup (Figure 2, Appendix A). The resulting phylogenetic tree was substantially validated by bootstrapping analyses.

The calculated mean divergence time for the order Polyporales is 854.52 million years ago (MYA), with a 95% highest posterior density (HPD) interval ranging from 754.59 to 955.66 MYA. The Polyporaceae family is estimated to have originated at a crown age of 620.39 MYA, within a 95% HPD interval of 471.72 to 847.65 MYA. The time of divergence between *D. qualens* and *C. qinlingensis* SNUT was determined to be 21.226 MYA, with a 95% HPD interval of 14.160 to 39.882 MYA. This divergence indicates a relatively recent evolutionary event within the Polyporaceae family (shown in Figure 2).

Further investigation using the reconstructed evolutionary trees revealed intricate patterns of gene contraction and expansion across 59,767 gene families in the genomes of these 31 species. Within the genome of *C. qinlingensis* SNUT, 113 out of 769 gene families were observed to have undergone expansion or contraction. In contrast, the genome of *D. squalens* showed corresponding variation in 212 out of 236 gene families. Notably, the genome of *C. qinlingensis* SNUT underwent more significant gene family contraction than that of *D. squalens* (Figure 2). This suggests that *C. qinlingensis* SNUT has experienced a significant reduction in gene families throughout its evolutionary history.

### 3.4. CAZyme Analysis

Cryptoporus species are the quintessential wood-decay fungi, predominantly found on the shady sides of dead pine wood or on the ground-facing surfaces of fallen logs. Although the mycelial growth and fruiting body formation of Cryptoporus species is closely linked to the activity of CAZymes, their CAZymes have not been studied in detail. A comprehensive analysis of the *C. qinlingensis* SNUT genome identified a total of 226 genes encoding 240 CAZyme domains (Appendix A, Appendix A). This diverse repertoire includes 109 glycoside hydrolases (GHs), 56 glycosyltransferases (GTs), 37 auxiliary activities (AAs), 23 carbohydrate esterases (CEs), 10 polysaccharide lyases (PLs) and five carbohydrate-binding modules (CBMs), as shown in Figure 3. It is note-worthy that 17 genes, exemplified by the gene YKJ-1005906, encode enzymes that possess dual CAZyme domains, highlighting the complexity and adaptability of these fungi in their ecological niche.

Of particular interest, 14 genes encode enzymes with dual CAZyme domains. In particular, the protein encoded by YKJ-1007594.1 contains two PL domains. The proteins translated by YKJ-1005906.1 and YKJ-1003713.1 each contain a pair of AA domains. Similarly, the proteins specified by YKJ-1004186.1 and YKJ-1007522.1 are characterized by two AA domains each, and four genes, including YKJ-1004186.1, encode proteins with a pair of CE domains. The protein encoded by YKJ-1001428.1 is characterized by a GH domain in addition to a GE domain, and a quartet of genes, including YKJ-1003717.1, encode proteins that integrate a GH domain with a CBM domain (Appendix A). The emergence of these multi-domain CAZymes underlines the complexity and adaptability of these fungi within their ecological niches. Although the clustering analysis indicates that *C. qinlingensis* SNUT forms a clade with members of the family Polyporaceae, the possession of a significantly higher number of GTs positions it as an outgroup within this branch. In terms of CAZyme profiles, *C. qinlingensis* SNUT was closest to *D. squalens* LYAD-421. (Figure 3).

### 3.5. Identifying Mating Genes and Developing SSR Markers

The tetrapolar mating system is the most extensive and complex sexual reproduction control system discovered so far in Basidiomycetes. Considering that the reason for the formation of the *Cryptoporus* fruiting body is still unknown and the cultivation demand is guided by its great medicinal value, it is necessary to analyze and identify its mating system. Upon examination of the haplotype A genome, it has been revealed that *C. qinlingensis* SNUT possesses a tetrapolar mating system, which is orchestrated by two unlinked mating-type (MAT) loci, namely MAT-A and MAT-B. The MAT-A locus of *C. qinlingensis* SNUT is situated on Chr 1 and encompasses a suite of coding genes. These include a gene for a mitochondrial intermediate peptidase (mip, YKJ-1000521.1), a gene for a HD transcription factor (HD, YKJ-1000522.1), a gene encoding an enigmatic conserved fungal protein (βfg, YKJ-1000523.1), and a gene for a glycosyltransferase family 8 protein (glgen, YKJ-1000525.1) (Figure 4A). As for the MAT-B genes, an array of five potential pheromone precursors and receptor genes have been identified, forming a cluster on Chr 7. Within this cluster, YKJ-1000521.1 to YKJ-1000523.1 represent three adjacent STE3s, while YKJ-1004917.1 and YKJ-1004927.1 are two separate STE3s which are not contiguous and are found dispersed at opposite ends of the region (Figure 4A).

Repeat sequences are an essential component of the genome, accounting for a significant proportion of the genome (Appendix A). Molecular tools based on these repetitive sequences, particularly microsatellite DNAs, play a crucial role in genetic breeding, variety identification and genetic map marker applications. In the haplotype A genome, a total of 11,665 dispersed repetitive sequence units were predicted, representing 26.64% of the total genome, with a total length of 9,206,229 bp. These dispersed repetitive sequences, also known as transposable elements (TEs), include four types in particular: short interspersed nuclear elements (SINEs), long interspersed nuclear elements (LINEs), long terminal repeats (LTRs), and DNA transposable elements (DNA-TEs). Among these, LTRs are the most abundant, accounting for 16.51% of the total genome. In addition, there are 6966 unknown types of repetitive sequences, accounting for 8.23% of the total genome (Appendix A). Further analysis revealed that the haplotype A genome contains 5970 tandem repetitive sequence units, including 4233 minisatellite DNAs and 770 microsatellite DNAs (Appendix A). Microsatellite DNAs, also known as simple sequence repeats (SSRs), are characterized by short repeat unit lengths, typically two to six nucleotides. They are widely distributed across the genome and exhibit a high degree of variability, making them valuable genetic markers for population genetic studies, paternity testing, and association analysis for genetic diseases. Using de novo searches, we analyzed the SSRs in the *C. qinlingensis* SNUT genome sequence with repeat unit lengths ranging from one to six nucleotides. A total of 1166 SSRs were identified in the 34.56 Mbp haplotype A genome and 1168 SSRs were found in the 34.38 Mbp haplotype A genome. The relative frequency of SSRs is about 34 per Mb (Appendix A). In haplotype A, the total length of SSRs is approximately 25.23 kb, representing 0.073% of the total length of the genome, while in haplotype B, the total length of SSRs is approximately 25.34 kb, also representing 0.073% of the total length of the genome.

In both the A haplotype and B haplotype, tri-nucleotide SSRs were the most common repeat types, accounting for 49.23% of all SSRs, followed by di-nucleotide repeats with 27.70% and hexa-nucleotide repeats with 12.09%. In the case of tri-nucleotide SSRs, the 14-fold ATA repeat formed the longest tri-nucleotide SSR, while the AAA, AXD, ACT, ADC, and TCA motifs were conspicuously absent. Among species with relatively close phylogenetic relationships, *D. squalens*, *Ganoderma sinense*, *Lentinus tigrinus*, and *Polyporus arcularius*, the number of tri-nucleotide SSRs was the highest, significantly outweighing the combined count of di- and mono-nucleotide SSRs that followed. In contrast, within another species with a relatively close phylogenetic affinity, *Trametes gibbosa*, the number of tri-nucleotide SSRs was slightly higher than that of di-nucleotide SSRs, yet the proportion of tri-nucleotide SSRs was considerably less than the combined proportion of di- and mono-nucleotide SSRs (Figure 4B).

### 3.6. Analysis and Identification of Transcription Factors and P450 Genes in C. qinlingensis

Based on HMMER models, 409 transcription factor (TF) sequences were identified in the *C. qinlingensis* genome, classified as follows: bZIP (125), FTD (40), Gti1_Pac2 (23), HMG (28), HOX (23), HSF (5), MYB (34), HLH (17), STE12 (2), TFIIB (16), Velvet (10), WHTH (2), ZnF-C2H2 (1), and ZnF-Zn2_Cys6 (82) (Figure 4C and Appendix A). The MrcA and ZnF-GATA types were not predicted. The variation in numbers between different TF families may be related to the specific functions of each TF family. In Basidiomycota, members of the bZIP transcription factor family are involved in a variety of biological processes, including stress response, secondary metabolism, cell cycle and development, pathogenicity, sporulation, signal transduction and gene expression regulation [21]. It is speculated that the various bZIP TFs play an important role in the life cycle of *C. qinlingensis*.

Cytochrome P450 (CYP450) is a key enzyme in fungal primary and secondary metabolism, playing an important role in detoxification, cellulose degradation, and biosynthesis of secondary metabolites [22,23]. By performing a clustering analysis with representative basidiomycete cytochrome P450 proteins from the fungal cytochrome P450 database, we successfully typed 77 P450 proteins selected from the *C. qinlingensis* genome (Supporting File). The identified P450 proteins include 18 members of the CYP5144 family, 11 members of the CYP5035 family, and 9 members of the CYP5150 family. There are also three members each of the CYP5037, CYP512, and CYP5113 families; two members each of the CYP5139 and CYP537 families; and one member each of the CYP502, CYP51, CYP5137, CYP5138, CYP5140, CYP5141, CYP5151, CYP5156, CYP53, CYP60, and CYP63 families. There are also seven CYP450s that have not yet been identified (Figure 5). Fungal CYP5144, a large class of heme thiolate proteins found in all organisms, plays a crucial role in many redox reactions within organisms [24]. It is speculated that these 18 CYP5144 family members may be closely associated with various oxidative reactions in *C. qinlingensis*.

### 3.7. Search and Analysis of Genes (Clusters) Involved in Secondary Metabolites

Given the traditional medicinal value of *C. qinlingensis* and the wealth of bioactive secondary metabolites characteristic of the *Cryptoporus* genus, a genomic search and analysis for genes (clusters) related to secondary metabolite biosynthesis have been conducted. Using the predictive capabilities of antiSMASH, a total of 30 BGCs involved in secondary metabolism were identified within the A haplotype genome of *C. qinlingensis* SNUT (Table 1). These gene clusters encode 37 core genes, including 19 terpene synthesis-related enzymes (Appendix A), nine NRPS-like enzymes, eight RiPP-like enzymes, and one each of β-lactone-related enzymes NRPS and PKS (Appendix A). These 30 BGCs are distributed over 13 chromosomes, with Chr 2A having the highest number of core genes with eight and Chr 13A having the lowest number with only one (Figure 6A).

Acknowledging the pivotal role of core genes in BGC-directed biosynthesis, an in-depth characterization of these genes ensued. YKJ-1001472.1, an NRPS, is predicted to contain seven domains (A-T-C-T-C-T-C). YKJ-1003785.1, defined by antiSMASH as an NRPS-like enzyme, was predicted by Clinker to be an NRPS containing four domains (C-A-T-R). Four of the seven NRPS-like enzymes contain the A-T-TR domains (YKJ-1004882.1, YKJ-1004672.1, YKJ-1003580.1 and YKJ-1003571.1). YKJ-1007479.1 and YKJ-1001927.1 each contain only A and T domains, whereas YKJ-1004687.1 contains A and TR domains (Figure 6B). YKJ-1003626.1 is the only PKS identified in this genome that shares 68.58% identity with its homologous PKS TBU27328.1 in *D. squalens*, and both share the same KS-AT-PT-ACP-TE domains (Figure 6C, Appendix A). Considering that drimane sesquiterpenes and their polymers are the main active constituents of *Cryptoporus* species, the terpene-related enzymes in the BGCs of *C. qinlingensis* were also analyzed (Appendix A). Among the 19 predicted terpene synthases, there are 17 sesquiterpene synthases (STSs), one squalene synthase (SQS), and one phytoene synthase (PSY) (Appendix A). To further perform a clustering analysis of the *C. qinlingensis*-derived STSs and infer their catalytic cyclization patterns of farnesyl pyrophosphate, an evolutionary tree-based clustering analysis was performed that included these 17 STSs along with 464 biochemically characterized basidiomycete-derived STSs. The evolutionary tree of basidiomycete STSs unambiguously identified 17 STSs in four distinct clades. Clade IV, characterized by the 1,6-cyclization of (3*R*, 5*S*)-nerolidyl diphosphate (NPP), contains nine STSs, whereas clade I contains four STSs and clades II and III contain two STSs each.

Among the 30 BGCs identified, three BGCs exhibit a 100% sequence similarity to known BGCs. Specifically, clusters 3 and 29 are found to be identical to the BGC associated with the biosynthesis of (+)-δ-cadinol, while cluster 4 matches the BGC responsible for the production of basidioferrin. Additionally, cluster 20 shares a 40% sequence similarity to the BGC involved in the biosynthesis of (+)-δ-cadinol, and cluster 23 shows a 22% sequence similarity to the BGC associated with the biosynthesis of ((−)-(R)-nephthenol.

## 4. Discussion

The Qinling Mountain Range, serving as the geographical demarcation line between the north and south of Western China [25], boasts a unique location and a complex climate that have given rise to a rich array of large fungal resources. These resources not only provide precious food and medicinal materials for the indigenous inhabitants of the southern Shaanxi Qinling Mountains but also hold a significant place in local traditional diets and medical practices. For instance, the wild *Hericium rajendrae*, discovered in Zhashui County, is a rare strain of lion’s mane mushroom that has long been consumed by the local population as a wild edible fungus. In Ankang City, *Laetiporus sulphureus* NWAFU-1, isolated on the trunk of sumac, is a wild edible and medicinal fungus with considerable value for both consumption and medicinal purposes. Meanwhile, in Zhenba County, *Cyathus olla* UST1 [26] is one of the earliest discovered medicinal fungi, traditionally used by the locals to alleviate headaches. As we continue to explore the large fungal resources of the Qinling region, *C. qinlingensis* SNUT has garnered attention due to its distinctive fruiting body morphology (Figure 1A). Preliminary research, based on morphological characteristics and multi-gene sequence alignment, has identified it as a new member of the *Cryptoporus* genus, marking the third species after *C. volvatus* and *C. sinensis*. In fact, the Qinling area is home to many unidentified new species of basidiomycetes, such as *Helvella longipes* [27] and *Hygrophoropsis ningshanica* [28], which are recent discoveries among the larger fungi in the region. In brief, the Qinling Mountains are not only a geographical boundary but also an underexplored and underutilized treasure trove of large fungal resources.

The Polyporaceae, a prominent family within the Basidiomycota, play a key role as decomposers in forest ecosystems, mainly parasitizing decaying dead wood. A few species, such as those in the genus *Cryptoporus*, are known to parasitize living trees. Despite the importance of the Polyporaceae in terms of edibility, medicinal properties, and ecological impact, and the fact that several of its members have had their genomes sequenced [20], the genomic landscape of *Cryptoporus* has remained largely unexplored. This research marks the first comprehensive genomic sequencing, chromosome-level, and haplotype-resolved assembly of *Cryptoporus qinlingensis*, facilitated by the Illumina NovaSeq sequencing platform, PacBio HiFi sequencing technology, and Hi-C sequencing technology. This achievement represents the first report of a genome within the genus *Cryptoporus*. Additionally, the realization of haplotype-resolved genome assembly holds significant importance for understanding the differences between the two sets of chromosomes in *C. qinlingensis*. *Dichomitus squalens* is a species with the closest genetic relationship to *C. qinlingensis* among those sequenced. The genomic sequencing of *C. qinlingensis* has revealed clear differences at the genomic level between these two species. Although their divergence is estimated to be around 212.26 MYAs, they share up to 6240 orthologous single-copy genes found in their genomes. In addition, analysis and identification of CAZyme, mating loci, SSR, TF, and P450 enriched the genome of *C. qinlingensis*. The sequencing of the *C. qinlingensis* genome has contributed significantly to our understanding of the genomic characteristics of the genus *Cryptoporus* and, more broadly, the family Polyporaceae. Furthermore, the revelation of the genomic characteristics of *C. qinlingensis* SNUT, as a wood-decay fungus and a folk medicinal mushroom, has propelled its role in cellulose degradation and pharmaceutical research. This will further highlight its potential applications in ecological cycling and health care.

In the field of medicinal fungi, secondary metabolites often play a crucial role as active components [29,30,31]. Recognizing this, we have characterized the secondary metabolite potential of *C. qinlingensis* based on genomic information. In the A haplotype, we predicted 30 secondary metabolite gene clusters, which collectively carry 37 core genes, including PKS, NRPS, NRPS-like, RIPP-like, and terpenes, indicating their ability to encode a diverse array of secondary metabolites. Typically, certain natural products in fungi exhibit species-specific characteristics. For example, cyathane diterpenoids are representative active constituents of the medicinal mushrooms *Cyathus* spp. [32] and *Hericium* spp. [33]. Styrylpyrone compounds, a class of high-value phenols with medicinal potential, are characteristic medicinal constituents produced by the well-known medicinal fungi genera *Phellinus* and *Inonotus* [34,35]. Drimane sesquiterpenoids are active components discovered in the genus *Cryptoporus* [7,36,37]. Although the secondary metabolites of *C. qinlingensis* have not been extensively investigated, it is speculated that drimane sesquiterpenoids may also be a major group of secondary metabolites in *C. qinlingensis*, with their synthetic enzymes being one of the 17 STSs. Although *C. qinlingensis* has a long history as a folk remedy in the region, the active components responsible for its medicinal effects remain unknown. This genomic-based analysis of secondary metabolic potential, coupled with subsequent metabolomic detection, will lay a solid foundation for the exploration of active components in this medicinal fungus.

## 5. Conclusions

This work has conducted genomic sequencing and de novo assembly of the newly identified medicinal fungus *C. qinlingensis*, with a sequencing depth exceeding 100×. The chromosome-level genome assembly and haplotype phasing have unveiled a high-quality and well-structured genome. The telomeric motifs of most chromosomes within the haplotypes have been identified. A phylogenetic analysis indicates that the genomic divergence between *C. qinlingensis* and *D. squalens* occurred approximately 212.26 MYAs. This recent divergence highlights the dynamic nature of fungal evolution and the rapid diversification within the Polyporaceae family. Comparative genomic analysis based on orthologous single-copy genes suggests that *C. qinlingensis* has experienced significant gene family contraction. On the other hand, the high-quality genome has facilitated the analysis and identification of the tetrapolar mating system, CAZymes, and P450 genes in *C. qinlingensis*. These features reveal the complexity of the genome. The exploration of BGCs for secondary metabolites has opened new avenues for the study of bioactive compounds in *C. qinlingensis*. A total of 30 BGCs were predicted in its haplotype A genome, including those encoding PKS, NRPS, NRPS-like, RIPP-like, and terpene synthesis-related enzymes, indicating its rich potential for producing a diverse array of secondary metabolites. The identification of its secondary metabolite biosynthetic potential provides important clues for the exploration of active components in this medicinal fungus. In summary, the genomic analysis of *C. qinlingensis* has greatly enhanced our understanding of this species and its genus. The findings of this study will serve as a valuable resource for future research aimed at harnessing the medicinal potential of *C. qinlingensis*, understanding its ecological roles, and exploring its evolutionary relationships within the Polyporaceae family. As we continue to unravel the genomic traits of this fascinating fungus, we can anticipate further discoveries that will contribute to the development of novel therapeutic agents and a deeper understanding of fungal biology and ecology.

## Figures and Tables

**Figure 1 jof-11-00163-f001:**
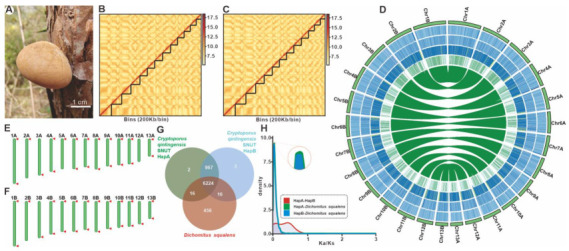
Morphological and genomic features and comparative genomic analyses of *C. qinlingensis* SNUT. (**A**) Morphological photographs of *C. qinlingensis* parasitizing pine trees. (**B**,**C**) Hi-C-based contig anchoring. The heat map shows the density of Hi-C interactions. (**D**) The genomic features of *C. qinlingensis* SNUT, from the outside to the inside are as follows: I. Chromosome; II–IV. GC-density, GC-skew, AT-skew (window size 1 kb); V. Gene-density (window size 1 kb). The central part of the diagram illustrates the collinearity between corresponding haplotypes. (**E**,**F**) Schematic representation of the telomeric motif on the assembled chromosomes of each haplotype. (**G**) Venn schematic of homologous gene comparison. (**H**) Ka/Ks comparison of *Cryptoporus* species and *D. squalens*.

**Figure 2 jof-11-00163-f002:**
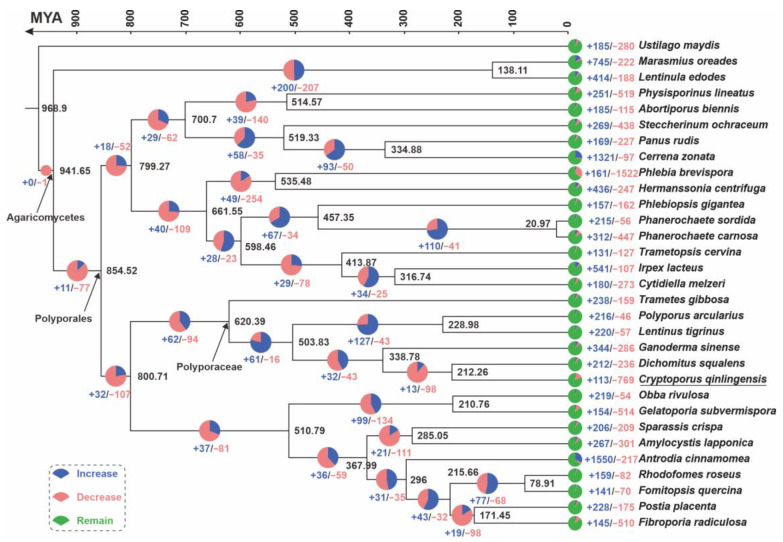
Phylogenetic insights and gene family dynamics. This figure illustrates the evolutionary relationships and fluctuations in gene family sizes within Cryptoporus species, contrasted with Polyporales species. The phylogenetic tree, constructed by maximum likelihood from 244 single-copy orthologous genes, enjoys robust bootstrap support at each node, with a 95% HPD. The mean crown age is given for each node, representing the estimated divergence times in MYA, which are also indicated by the colored numbers adjacent to the branches. The relative amount of genomic expansion and contraction for each species is shown beside its name.

**Figure 3 jof-11-00163-f003:**
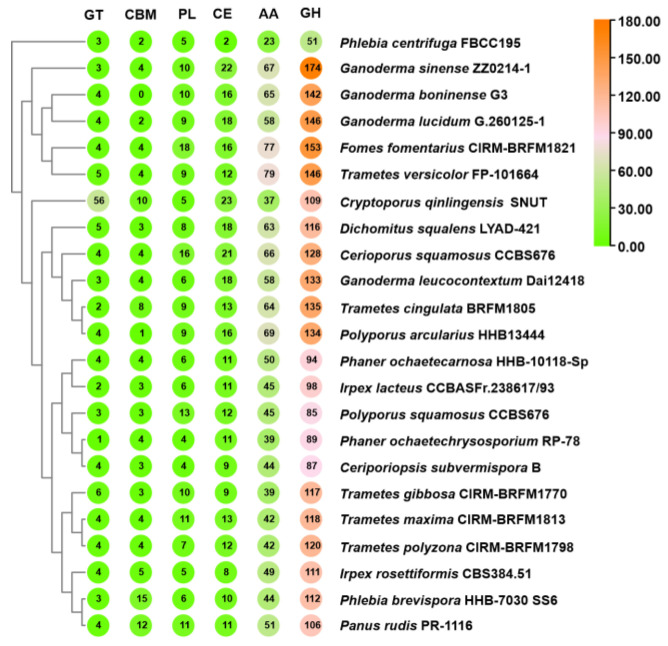
CAZymes analysis of *C. qinlingensis* SNUT and related wood-decay fungi. GT, CBM, PL, CE, AA, and GH refer to glycosyltransferase, carbohydrate-binding module, polysaccharide lyase, carbohydrate esterase, auxiliary activity, and glycoside hydrolase, respectively. The sizes and colors (from green through pink to yellow) of the circles indicate the change in quantity.

**Figure 4 jof-11-00163-f004:**
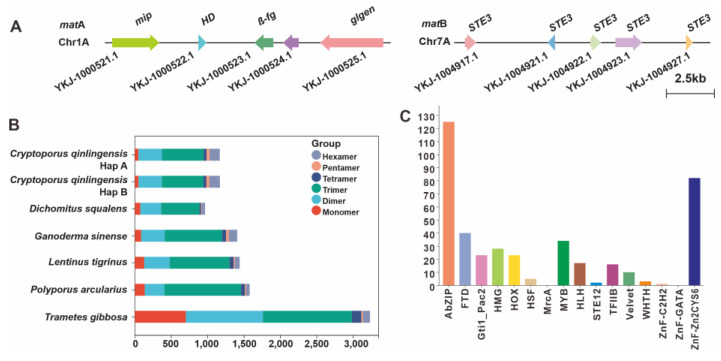
Identification of the mating genes and comparison of SSR abundance of *Cryptoporus* species. (**A**) Structural diagram of the genes on the matA locus and matB locus of *C. qinlingensis* SNUT. (**B**) The relative abundance of SSRs in the genome of *C. qinlingensis* SNUT and its five relatively closely related macrofungi. (**C**) The abundance of different types of transcription factors in the genome of *C. qinlingensis* SNUT.

**Figure 5 jof-11-00163-f005:**
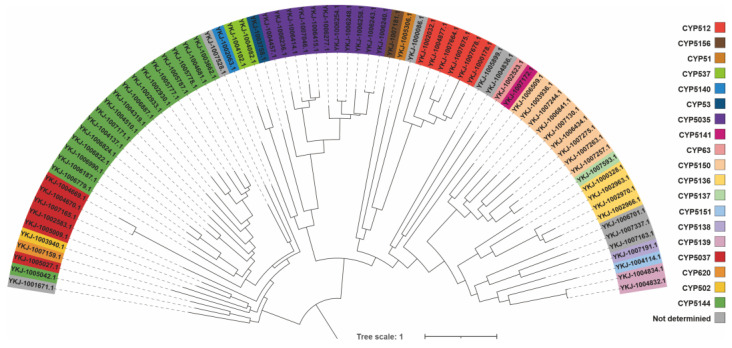
Cluster analysis of 77 P450s from *C. qinlingensis* based on the maximum-likelihood tree. Each P450 family is shown in a separate color.

**Figure 6 jof-11-00163-f006:**
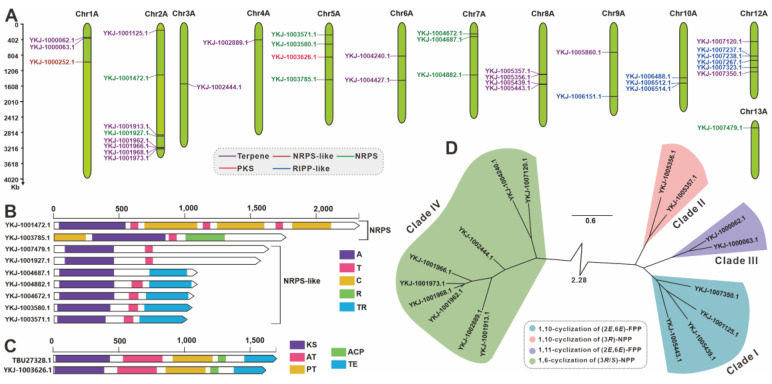
Analysis of genes involved in secondary-metabolite biosynthesis. (**A**) Distribution of biosynthetic core genes for natural products on the chromosomes. (**B**,**C**) Domain analyses of NRPS, NRPS-like, and PKS. (**D**) Phylogenetic tree analysis for STSs.

**Table 1 jof-11-00163-t001:** Putative biosynthetic gene clusters responsible for secondary metabolites in the genome of *C. qinlingensis* SNUT.

Cluster No.	Location	Start (bp)	End (bp)	Core Gene ID	Core Gene Type
1	Chr1A	359,344	398,787	YKJ-1000062.1	Terpene
2	Chr1A	999,437	1,046,045	YKJ-1000252.1	β-lactone
				YKJ-1000255.1	NRPS-like
3	Chr2A	174,241	195,721	YKJ-1001125.1	Terpene
4	Chr2A	2,883,761	2,905,036	YKJ-1001913.1	Terpene
5	Chr2A	3,211,700	3,266,305	YKJ-1001962.1	Terpene
6	Chr2A	1,315,680	1,363,525	YKJ-1001472.1	NRPS
7	Chr2A	2,907,662	2,952,731	YKJ-1001927.1	NRPS-like
8	Chr3A	1,573,377	1,594,751	YKJ-1002444.1	Terpene
9	Chr4A	443,477	464,752	YKJ-1002889.1	Terpene
10	Chr5A	288,054	332,091	YKJ-1003571.1	NRPS-like
11	Chr5A	530,357	574,388	YKJ-1003580.1	NRPS-like
12	Chr5A	1,444,291	1,491,423	YKJ-1003785.1	NRPS
13	Chr5A	862,970	908,215	YKJ-1003626.1	PKS
14	Chr6A	855,429	876,688	YKJ-1004240.1	Terpene
15	Chr6A	1,486,316	1,507,452	YKJ-1004427.1	Terpene
16	Chr7A	272,998	317,249	YKJ-1004672.1	NRPS-like
17	Chr7A	340,327	384,582	YKJ-1004687.1	NRPS-like
18	Chr7A	1,337,635	1,381,910	YKJ-1004882.1	NRPS-like
19	Chr8A	1,324,708	1,349,324	YKJ-1005356.1	Terpene
				YKJ-1005357.1	Terpene
20	Chr8A	1,567,264	1,600,522	YKJ-1005439.1	Terpene
				YKJ-1005443.1	Terpene
21	Chr9A	761,639	783,524	YKJ-1005860.1	Terpene
22	Chr9A	1,894,286	1,954,964	YKJ-1006151.1	RiPP
23	Chr10A	1,393,108	1,453,799	YKJ-1006488.1	RiPP
24	Chr10A	1,533,952	1,597,764	YKJ-1006512.1	RiPP
				YKJ-1006514.1	RiPP
25	Chr12A	498,674	520,051	YKJ-1007120.1	Terpene
26	Chr12A	1,271,344	1,292,712	YKJ-1007350.1	Terpene
27	Chr12A	849,229	911,262	YKJ-1007237.1	RiPP
				YKJ-1007238.1	RiPP
28	Chr12A	957,737	1,018,469	YKJ-1007267.1	RiPP
29	Chr12A	1,147,393	1,208,064	YKJ-1007323.1	RiPP
30	Chr13A	155,448	200,573	YKJ-1007479.1	NRPS-like

Chr indicates chromosome.

## Data Availability

The original contributions presented in this study are included in the article/Appendix A. Further inquiries can be directed to the corresponding authors.

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
