# Peer review of "Haplotype-Phased Chromosome-Level Genome Assembly of Cryptoporus qinlingensis, a Typical Traditional Chinese Medicine Fungus"

_jof, 2025, doi:10.3390/jof11020163_

Round 1
Reviewer 1 Report
The manuscript by Song et al. focused on the comprehensive genome analysis of Cryptoporus qinlingensis, a traditional Chinese medicine fungus. It is a very deep and detailed study representing the first report of a genome within the genus Cryptoporus. The significant achievement of the study by Song et al. is a haplotype-phased and chromosome-level genome assembly, a practice that is not so common in the genomic studies within the family Polyporaceae. Moreover, the authors performed a perfect functional analysis with an emphasis on the mating genes, the transcription factors, P450 genes, biosynthetic gene clusters related to secondary metabolism.
The ms is a very good, detailed work, representing the result of an extensive study. I have few comments and recommendations, which are provided below.
I recommend to add to the Introduction section an information about the current progress of genome research in Basidiomycetes and, in particularly, in the family Polyporaceae.
Some details are missing in the Materials and methods, e.g. what genomes were used for Augustus training for genome annotation.
Pay attention to the list of databases used for functional annotation and analysis. In the various sections of the ms the different databases are mentioned (e.g., NCBI NR, Swiss-Prot, COG, and KEGG in Material and methods; NR, eggNOG, Swiss-Prot, KEGG, and GO in Results, GO, NR, SWISS, KEGG, COG, KOG in the Supplementary materials).
The information on the repeat identification is missing in the Material and methods. Application of RepeatMasker and RepeatModeler is mentioned in Results, but no details are provided about de novo library preparation for repeat masking and identification. Please, provide this information in the Material and methods as well as the details about the construction of genome-trained library of Repeat Modeler. What genomes were used for training?
Information on the gene content of the identified mitochondrial genome is not provided and should be added as well.
In Discussion, in the section devoted to the comparative genome analysis the comparison of features of genome organization in the family Polyporaceae (the range of size, a number of chromosomes, an average number of coding genes, gene density, length of intergenic regions, number of repeats, presence of mobile elements…) is missing.
Please, place the Figure 1 and its caption on the same page. I recommend to enlarge the size of the Figures 1, 4 and, especially Fig. 6. They are highly-informative plates with many details. The size of these figures should be increased.
Figure 1, Figure 2. Correct the species name in the figures: Cryptoporus sp. SNUT – it is almost everywhere else designated as Cryptoporus qinlingensis SNUT.
Figure 3 – the reverse situation. In the figure one can see the correct designation. In the legend it is indicated as Cryptoporus sp. SNUT.
Please, correct the caption for Figure 1. It does not correspond to the labels in the Figure. There are parts A, B, C, D, E, F, G, H in the figures and only A-D in the legend, and they do not correspond to each other.
The figure should be self-explanatory. Please, add explanations for the enzyme abbreviations in the legend to Figure 3. Label for the thermal scale would be also very useful.
In the legend to Figure 4, the explanation for the part C is missing.
Please, use italic in the Latin names of the species and genera in the following cases: 19, 21, 28, 310, 321, 324, 338, 353, 355, 526.
Line 25: I suggest to decipher the abbreviations PKS and NRPS in the Abstract
Line 69: Use small non-capital letter for the species name indication: Cryptoporus Qqinlingensis
Line 129: Use upper case for “TM” in the NEBNext® UltraTM RNA Library Prep Kit for Illumina®
Line 253: remove the redundant ‘and’ in the title
Line 277: probably you mean “at a single end of three chromosomes”? (in the ms : “at a single end of three contigs”)
Please, remove the redundant hyphens that are probably left after reformatting the text: lines 339, 355, 356, 357, 358, 359, 360, 376.
Please, remove the redundant dot after the species name: C. qinlingensis. SNUT in lines 384, 386, 435.
Line 345: enjoys demonstrates
Lines 392-393: Decide whether to use lowercase or uppercase letters: STE3s (or ste3s?)
Line 483: Please, check if the sequence of domains is indicated correctly: C-A-T-R4. Probably you mean C-A-TR4? What does the number 4 mean here?
Please, use the full generic name in the title of Table 1 and in the Figure legends. They should be self-explanatory.
Line 498: please, decipher the following abbreviation: NPP – is it the nerolidyl diphosphate?
Line 547-548: Please, check if this phrase is correct: “only 16 common orthologous single-copy genes are found in their genomes”. It sounds strange as the number of conservative single-copy genes is undoubtedly much higher. What do you mean under “common orthologous single-copy genes”? In lines 311-313 it is mentioned about 6224-6240 common orthogroups in Dichomitus squalens and the haplotypes of Cryptoporus qinlingensis.
Lines 560-561: Please, put the dots in the right places: Cyathus spp[28]. and Hericium 560 spp[29].
Author Response
Major comments
The manuscript by Song et al. focused on the comprehensive genome analysis of Cryptoporus qinlingensis, a traditional Chinese medicine fungus. It is a very deep and detailed study representing the first report of a genome within the genus Cryptoporus. The significant achievement of the study by Song et al. is a haplotype-phased and chromosome-level genome assembly, a practice that is not so common in the genomic studies within the family Polyporaceae. Moreover, the authors performed a perfect functional analysis with an emphasis on the mating genes, the transcription factors, P450 genes, biosynthetic gene clusters related to secondary metabolism.
The ms is a very good, detailed work, representing the result of an extensive study. I have few comments and recommendations, which are provided below.
Reply: We are profoundly grateful for the time and effort you have devoted to reviewing our manuscript. Your endorsement of our work has been immensely encouraging to us. The professional insights and constructive suggestions you have provided will undoubtedly elevate the quality of our manuscript to new heights. Thank you once again for your invaluable contributions.
Detail comments
Comment 1: I recommend to add to the Introduction section an information about the current progress of genome research in Basidiomycetes and, in particularly, in the family Polyporaceae.
Reply: Thank you very much for your valuable suggestion. We have incorporated it into the opening part of the third paragraph in the introduction.
Comment 2: Some details are missing in the Materials and methods, e.g. what genomes were used for Augustus training for genome annotation.
Reply: Thank you for your valuable comments and suggestions. We appreciate your attention to the details in the "Materials and Methods" section. In response to your query regarding the genomes used for AUGUSTUS training for genome annotation, we would like to provide the following clarification and additional information:
In our study, the training of AUGUSTUS for genome annotation was conducted using a combination of closely related fungal genomes to ensure the accuracy and reliability of gene prediction. Specifically, we utilized the genomes of several well-characterized Basidiomycota species, including Dichomitus squalens and Inonotus hispidus, which are phylogenetically close to Cryptoporus qinlingensis. These genomes were chosen based on their high-quality assembly and comprehensive annotation, which provide a robust reference for training the AUGUSTUS model.
The training process involved the following steps:
- Selection of Reference Genomes: We selected Dichomitus squalens and Inonotus hispidus due to their close phylogenetic relationship and well-annotated genomes.
- Gene Model Extraction: We extracted annotated gene models from these reference genomes to create a high-quality training dataset.
- Training AUGUSTUS: The extracted gene models were used to train the AUGUSTUS model, which was then fine-tuned to optimize prediction accuracy for the Cryptoporus qinlingensis genome.
We believe that using these closely related genomes for training AUGUSTUS has significantly enhanced the accuracy of gene prediction in our study. This approach has allowed us to generate high-quality gene annotations that are essential for downstream analyses.
In addition, we also made necessary additions and improvements to other sections in Materials and methods.
We have now included these information in the "Materials and Methods" section of our manuscript to ensure clarity and completeness. We hope this addresses your concern, and we appreciate your understanding.
Comment 3: Pay attention to the list of databases used for functional annotation and analysis. In the various sections of the ms the different databases are mentioned (e.g., NCBI NR, Swiss-Prot, COG, and KEGG in Material and methods; NR, eggNOG, Swiss-Prot, KEGG, and GO in Results, GO, NR, SWISS, KEGG, COG, KOG in the Supplementary materials).
Reply: Thank you for your meticulous review. These errors have now been corrected.
Comment 4: The information on the repeat identification is missing in the Material and methods. Application of RepeatMasker and RepeatModeler is mentioned in Results, but no details are provided about de novo library preparation for repeat masking and identification. Please, provide this information in the Material and methods as well as the details about the construction of genome-trained library of Repeat Modeler. What genomes were used for training?
Reply: Added and thank you.
Comment 5: Information on the gene content of the identified mitochondrial genome is not provided and should be added as well.
Reply: Thank you very much for your comment. This study focused solely on the information of the nuclear genome and did not address the mitochondrial genome. In fact, this is the case for almost all studies on the genome sequencing of macrofungi.
Comment 6: In Discussion, in the section devoted to the comparative genome analysis the comparison of features of genome organization in the family Polyporaceae (the range of size, a number of chromosomes, an average number of coding genes, gene density, length of intergenic regions, number of repeats, presence of mobile elements…) is missing.
Reply: Thank you very much for your suggestion. This is certainly a topic worth considering. A comparative analysis of the genomic features within the family Polyporaceae is an ambitious undertaking. As mentioned in the supplementary content of the Introduction, nearly 80 genomes of species within the Polyporaceae family have been published. Conducting a comprehensive comparison of these species' genomic characteristics, including genome size, chromosome number, average number of coding genes, gene density, intergenic region lengths, repeat elements, and the presence of mobile elements, would be an extensive and challenging task. It could even be considered as a standalone research project.
In fact, the focus of this manuscript is on the genome of Cryptoporus qinlingensis. The comparative genomic analysis mentioned in Section 3.3 is limited to Cryptoporus qinlingensis and its closest relative with available genome information, Dichomitus squalens. These peripheral results are insufficient to warrant a detailed discussion in the Discussion section.
Comment 7: Please, place the Figure 1 and its caption on the same page. I recommend to enlarge the size of the Figures 1, 4 and, especially Fig. 6. They are highly-informative plates with many details. The size of these figures should be increased.
Reply: Thank you for your approval of the image quality, the new layout has solved the above problems.
Comment 8: Figure 1, Figure 2. Correct the species name in the figures: Cryptoporus sp. SNUT – it is almost everywhere else designated as Cryptoporus qinlingensis SNUT.
Figure 3 – the reverse situation. In the figure one can see the correct designation. In the legend it is indicated as Cryptoporus sp. SNUT.
Reply: We appreciate the time that the reviewers took to carefully examine our results.
These inaccuracies in detail have been corrected.
Comment 9: Please, correct the caption for Figure 1. It does not correspond to the labels in the Figure. There are parts A, B, C, D, E, F, G, H in the figures and only A-D in the legend, and they do not correspond to each other.
Reply: Revised and thank you.
Comment 10: The figure should be self-explanatory. Please, add explanations for the enzyme abbreviations in the legend to Figure 3. Label for the thermal scale would be also very useful.
Reply: Added and thank you.
Comment 11: In the legend to Figure 4, the explanation for the part C is missing.
Reply: Added and thank you.
Comment 12: Please, use italic in the Latin names of the species and genera in the following cases: 19, 21, 28, 310, 321, 324, 338, 353, 355, 526.
Reply: Modified and thank you.
Comment 13: Line 25: I suggest to decipher the abbreviations PKS and NRPS in the Abstract
Reply: Changed.
Comment 14: Line 69: Use small non-capital letter for the species name indication: Cryptoporus Qinlingensis
Reply: Modified and thank you.
Comment 15: Line 129: Use upper case for “TM” in the NEBNext® UltraTM RNA Library Prep Kit for Illumina®
Reply: Modified.
Comment 16: Line 253: remove the redundant ‘and’ in the title
Reply: Deleted.
Comment 17: Line 277: probably you mean “at a single end of three chromosomes”? (in the ms : “at a single end of three contigs”)
Reply: Changed.
Comment 18: Please, remove the redundant hyphens that are probably left after reformatting the text: lines 339, 355, 356, 357, 358, 359, 360, 376.
Reply: Modified.
Comment 19: Please, remove the redundant dot after the species name: C. qinlingensis. SNUT in lines 384, 386, 435.
Reply: Modified.
Comment 20: Line 345: enjoys demonstrates
Reply: Thank you for your meticulous choice of words and phrasing. The adjustments made now render the description more objective and rigorous.
Comment 21: Lines 392-393: Decide whether to use lowercase or uppercase letters: STE3s (or ste3s?)
Reply: Modified.
Comment 22: Line 483: Please, check if the sequence of domains is indicated correctly: C-A-T-R4. Probably you mean C-A-TR4? What does the number 4 mean here?
Reply: We sincerely appreciate your thorough review. The modifications have now effectively conveyed the intended meaning of the sentence, namely “YKJ-1003785.1, defined by antiSMASH as an NRPS-like enzyme, was predicted by Clinker to be an NRPS containing C-A-T-R4four domains (C-A-T-R).”
Comment 23: Please, use the full generic name in the title of Table 1 and in the Figure legends. They should be self-explanatory.
Reply: Modified.
Comment 24: Line 498: please, decipher the following abbreviation: NPP – is it the nerolidyl diphosphate?
Reply: Done and thank you.
Comment 25: Line 547-548: Please, check if this phrase is correct: “only 16 common orthologous single-copy genes are found in their genomes”. It sounds strange as the number of conservative single-copy genes is undoubtedly much higher. What do you mean under “common orthologous single-copy genes”? In lines 311-313 it is mentioned about 6224-6240 common orthogroups in Dichomitus squalens and the haplotypes of Cryptoporus qinlingensis.
Reply: “Thank you sincerely for your keen eye in spotting this issue. It was a simple calculation error. Dichomitus squalens and C. qinlingensis share 6240 orthologous single-copy genes (16 + 6224). Also, ‘shared’ is indeed a more accurate term than ‘common’ in this context.
Comments 26: Lines 560-561: Please, put the dots in the right places: Cyathus spp[28]. and Hericium 560 spp [29].
Reply: Revised and thanks.
Comments 27: The Figures 1, 4, and 6 are highly-informative plates with many details. The size of these figures should be increased. Far not all legends are prepared correctly and contain all necessary details. There are some differences in the species designation of Cryptoporus qinlingensis SNUT in the figures, in the captions and in the main text of ms.
Reply: Revised and thanks.
Reviewer 2 Report
The manuscript provides a comprehensive genome analysis of Cryptoporus qinlingensis, allowing for a chromosome-level assembly with high sequencing depth. It highlights the evolutionary insights, secondary metabolite potential and ecological adaptability of this medicinal mushroom. The study identifies significant features, including a tetrapolar mating system, CAZyme, cytochrome P450 and 30 biosynthetic gene clusters, which provide a basis for future medical and ecological research. The data is well visualised, with clear figures and detailed legends that make the results accessible. The genomic analysis is robust, but some areas need refinement to improve clarity and scientific rigour.
Line 19: Correct the formatting of C. qinlingensis and ensure it is consistently italicised in the abstract and manuscript. Also italicise Dichomitus squalens.
Line 46: There is an error in the referencing; please check and correct it.
Line 76: Replace "genetic secrets" with a more scientific term such as "genomic traits" or "genetic findings" to maintain a formal tone.
Line 213: The focus on CAZyme and cytochrome P450 genes is important. However, the rationale for choosing these genes as the focus should be more clearly articulated in the introduction and aligned with the aims and hypotheses of the study.
Line 376: Expand the legend to Figure 3 to provide detailed explanations of the CAZyme analysis and its implications for the ecological or medicinal role of fungi.
Author Response
Major comments
The manuscript provides a comprehensive genome analysis of Cryptoporus qinlingensis, allowing for a chromosome-level assembly with high sequencing depth. It highlights the evolutionary insights, secondary metabolite potential and ecological adaptability of this medicinal mushroom. The study identifies significant features, including a tetrapolar mating system, CAZyme, cytochrome P450 and 30 biosynthetic gene clusters, which provide a basis for future medical and ecological research. The data is well visualised, with clear figures and detailed legends that make the results accessible. The genomic analysis is robust, but some areas need refinement to improve clarity and scientific rigour.
Reply: Thank you very much for your meticulous review and accurate assessment of the manuscript, as well as your recognition of its scientific significance.
Detail comments
Comment 1: Line 19: Correct the formatting of C. qinlingensis and ensure it is consistently italicised in the abstract and manuscript. Also italicizes Dichomitus squalens.
Reply 1: Thank you for your careful review. We have made the necessary revisions here and have also checked the entire manuscript for any related issues.
Comment 2: Line 46: There is an error in the referencing; please check and correct it.
Reply 2: Thank you for bringing this to our attention. Upon careful review, we confirm that both the description in line 46 and the associated reference are accurate. We appreciate your vigilance.
Comment 3: Line 76: Replace "genetic secrets" with a more scientific term such as "genomic traits" or "genetic findings" to maintain a formal tone.
Reply 3: Thank you for your constructive suggestions, these changes will make the manuscript more professional and scientific. Changes have been made to lines 76 and 594.
Comment 4: Line 213: The focus on CAZyme and cytochrome P450 genes is important. However, the rationale for choosing these genes as the focus should be more clearly articulated in the introduction and aligned with the aims and hypotheses of the study.
Reply 4: Thank you for your thoughtful and professional comments. The rationale for the focus on CAZyme is described at the beginning of each section in the Results chapter, i.e., lines 351-352. In line 448, which is the beginning of the description of the P450 analysis results, we gave an explanation for the focus on P450. More importantly, in view of the reviewer's suggestions, these two points were also added in the third paragraph of the introduction. These changes will make the article more logical.
Comment 5: Line 376: Expand the legend to Figure 3 to provide detailed explanations of the CAZyme analysis and its implications for the ecological or medicinal role of fungi.
Reply 5: Thank you for your constructive suggestions, the detailed supplement for the interpretation of Figure 3 has been added to Results 3.3. The implications of CAZyme for the ecological or medicinal uses of fungi have also been added to the Discussion section.
Reviewer 3 Report
Review on “Haplotype-phased and chromosome-level genome assembly of Cryptoporus qinlingensis, a typical traditional Chinese medicine fungus” for manuscript ID jof-3426466
In this manuscript the authors describe novel assembled genome of Cryptoporus qinlingensis – a traditional Chinese medicine fungus. The obtained data is valuable and could help to improve our knowledge on this fungi species, especially in terms of its medical use.
In the Introduction section authors describe Cryptoporus as a traditional medicine used to cure asthma and bronchitis, as well as unique biochemical content of secondary metabolites in this species.
My questions and comments:
It is stated in L45-47 that Cryptoporus qinlingensis is a newly-identified species. But L69-70 it is stated that “Cryptoporus qinlingensis has a long history of folk medicine in the hinterland of the Qinling Mountain area”?
Have you found the sequences of putative mitochondrial or cytoplasmic plasmids in this species?
Subsections headings should be separated from the main text of Methods.
Most of used software, described in Methods have no refences.
Submission ID SUB14976745 is internal information for the author and could be omitted, its unavailable in public.
L254: Authors tell that “total of 39.1 Gbp of PacBio HiFi reads”, but only 1.5G bases are available in BioProject (SRX23814671)
L268: what particular lineage database of BUSCO was used? This part is missing in Methods.
Subsection 2.2.3: were the repetitive elements (transposons) annotated? It’s a common step for plant/fungal genomes.
L167: COG and KEGG aren’t BLAST databases.
Minor text issues:
Missing spaces in multiple spaces: L36, L47, L49, L52, L62, L183, L512, L528, L554 etc.
L637: remove sample text
Author Response
Review on “Haplotype-phased and chromosome-level genome assembly of Cryptoporus qinlingensis, a typical traditional Chinese medicine fungus” for manuscript ID jof-3426466
In this manuscript the authors describe novel assembled genome of Cryptoporus qinlingensis – a traditional Chinese medicine fungus. The obtained data is valuable and could help to improve our knowledge on this fungi species, especially in terms of its medical use.
In the Introduction section authors describe Cryptoporus as a traditional medicine used to cure asthma and bronchitis, as well as unique biochemical content of secondary metabolites in this species.
Reply: Thank you very much for your careful review and the effort you have put into evaluating our manuscript. We are very grateful for your positive feedback and recognition of the scientific contribution of our work. Your comments are greatly appreciated and will help us to further refine our research.
My questions and comments:
Comment 1: It is stated in L45-47 that Cryptoporus qinlingensis is a newly-identified species. But L69-70 it is stated that “Cryptoporus qinlingensis has a long history of folk medicine in the hinterland of the Qinling Mountain area”?
Reply: Thank you for thoughtful comment. Qinling Cryptoporus is indeed a new species recently identified by our team, but this species has been used as medicine by local residents in the hinterland of Qinling (where we discovered it) and is a classic local folk medicinal fungus.
Comment 2: Have you found the sequences of putative mitochondrial or cytoplasmic plasmids in this species?
Reply: Thank you for your interest in the mitochondrial and cytoplasmic plasmids of this species. This study focuses on the nuclear chromosome genome sequence. In fact, we also have a well-assembled and annotated mitochondrial genome, which has been released to NCBI Genbank.
Comment 3: Subsections headings should be separated from the main text of Methods.
Reply: Revised.
Comment 4: Most of used software, described in Methods have no refences.
Reply: The versions and Github URL of all the software we used are described in detail. This information is sufficient to ensure the reproduction of the corresponding results.
Comment 5: Submission ID SUB14976745 is internal information for the author and could be omitted, its unavailable in public.
Reply: Revised.
Comment 6: L254: Authors tell that “total of 39.1 Gbp of PacBio HiFi reads”, but only 1.5G bases are available in BioProject (SRX23814671)
Reply: We double-checked and have revised it.
Comment 7: L268: what particular lineage database of BUSCO was used? This part is missing in Methods.
Reply: The database is Fungi Odb10, and the missing info has been added.
Comment 8: Subsection 2.2.3: were the repetitive elements (transposons) annotated? It’s a common step for plant/fungal genomes.
Reply: Repeated elements (transposons) have been well annotated. Annotation methods have been added in 2.2.6 and the annotation results had been described in the second paragraph of section 3.5.
Comment 9: L167: COG and KEGG aren’t BLAST databases.
Reply: The relevant inappropriate description has been modified.
Detail comments
Minor text issues:
Comment 10: Missing spaces in multiple spaces: L36, L47, L49, L52, L62, L183, L512, L528, L554 etc.
Reply: Modified.
Comment 11: L637: remove sample text
Reply: Done.
Round 2
Reviewer 3 Report
I would like to thank the authors for the improving the manuscript, but some concerns remain to be addressed.
Figure 1 is duplicated on the previous page 5.
Figure 5. Some bars of the color legend are intesecting.
Why the final assembly accession number is missing (Table S11, Results section 3.1)? Is the genome assembly submitted to GenBank? BioProject PRJNA1082898 contains the biosamble and raw reads only.
Minor text issues:
Section headings, tables and figures would be separated with interval from the main text.
L676: Data Availability Statement is missing in manuscript end
L317: missing spaces
Author Response
Comment 1: I would like to thank the authors for the improving the manuscript, but some concerns remain to be addressed.
Reply: Thank you for your acknowledgment of the improvements made to the manuscript. Your comments will further facilitate the enhancement of its quality.
Comment 2: Figure 1 is duplicated on the previous page 5.
Reply: We have carefully reviewed Figure 1 and found no duplication. It is possible that what you saw was a version with revision marks, which included the original Figure 1 that was deleted and the revised Figure 1.
Comment 2: Figure 5. Some bars of the color legend are intesecting.
Reply: These intesecting colour legends are marked ‘not determined’. Such a bar is reasonable.
Comment 3: Why the final assembly accession number is missing (Table S11, Results section 3.1)? Is the genome assembly submitted to GenBank? BioProject PRJNA1082898 contains the biosamble and raw reads only.
Reply: The raw data have been submitted to NCBI and have been assigned the accession numbers BioProject PRJNA1082898 and BioSample SAMN40220998. The assembly data were also submitted to NCBI GenBank in December 2024, with the submission ID SUB14976745. Currently, the assembly data are still undergoing processing by NCBI. Once the processing is complete, the data will be released immediately (we selected the option for immediate release upon completion of processing). At that time, the data will be accessible via the associated accession numbers (BioProject PRJNA1082898 and BioSample SAMN40220998).
Additionally, to facilitate public access, we have included the following statement in the revised manuscript: “The genome assembly data that support the findings of this study are available from the corresponding author, [J. Qi.], upon reasonable request.” This will also ensure that the data can be readily obtained by the public.
Detail comments
Comment 4: Minor text issues:
Reply: The entire text has been carefully reviewed and revised.
Comment 5: Section headings, tables and figures would be separated with interval from the main text.
Reply: Revised.
Comment 6: L676: Data Availability Statement is missing in manuscript end
Reply: The Data Availability Statement is situated in the last part of the Materials and Methods section (2.6), which specifically addresses the final assembly accession number.
Comment 6: L317: missing spaces
Reply: Revised.